# Investigation of the Impact of Point Defects in InGaN/GaN Quantum Wells with High Dislocation Densities

**DOI:** 10.3390/nano13182569

**Published:** 2023-09-16

**Authors:** Pierre Lottigier, Davide Maria Di Paola, Duncan T. L. Alexander, Thomas F. K. Weatherley, Pablo Sáenz de Santa María Modroño, Danxuan Chen, Gwénolé Jacopin, Jean-François Carlin, Raphaël Butté, Nicolas Grandjean

**Affiliations:** 1Advanced Semiconductors for Photonics and Electronics Laboratory, Institute of Physics, École Polytechnique Fédérale de Lausanne (EPFL), 1015 Lausanne, Switzerland; davidemaria.dipaola@you.unipa.it (D.M.D.P.); thomas.weatherley@epfl.ch (T.F.K.W.); danxuan.chen@epfl.ch (D.C.); jean-francois.carlin@epfl.ch (J.-F.C.); raphael.butte@epfl.ch (R.B.); nicolas.grandjean@epfl.ch (N.G.); 2Electron Spectrometry and Microscopy Laboratory, Institute of Physics, École Polytechnique Fédérale de Lausanne (EPFL), 1015 Lausanne, Switzerland; duncan.alexander@epfl.ch; 3Université Grenoble Alpes, CNRS, Grenoble INP, Institut Néel, 38000 Grenoble, France; pablo.saenz-de-santa-maria-modrono@neel.cnrs.fr (P.S.d.S.M.M.); gwenole.jacopin@neel.cnrs.fr (G.J.)

**Keywords:** quantum well, point defect, threading dislocation, photoluminescence, cathodoluminescence, electron microscopy

## Abstract

In this work, we report on the efficiency of single InGaN/GaN quantum wells (QWs) grown on thin (<1 µm) GaN buffer layers on silicon (111) substrates exhibiting very high threading dislocation (TD) densities. Despite this high defect density, we show that QW emission efficiency significantly increases upon the insertion of an In-containing underlayer, whose role is to prevent the introduction of point defects during the growth of InGaN QWs. Hence, we demonstrate that point defects play a key role in limiting InGaN QW efficiency, even in samples where their density (2–3 × 10${}^{9}$ cm${}^{-2}$) is much lower than that of TD (2–3 × 10${}^{10}$ cm${}^{-2}$). Time-resolved photoluminescence and cathodoluminescence studies confirm the prevalence of point defects over TDs in QW efficiency. Interestingly, TD terminations lead to the formation of independent domains for carriers, thanks to V-pits and step bunching phenomena.

## 1. Introduction

Since the first report of candela-class blue light-emitting diodes (LEDs) [1], efficient light emission has been reported from InGaN/GaN quantum wells (QWs) heteroepitaxially grown on *c*-plane sapphire substrates, despite threading dislocation (TD) densities $>10^{9}$ cm${}^{-2}$ [2,3,4]. By contrast, a dislocation density higher than $10^{3}$ cm${}^{-2}$ causes a tenfold drop in efficiency in red-emitting AlGaInP-based LEDs [3] and GaAs-based devices [5,6]. Building upon the observation of such high-efficiency III-nitride (III-N) QWs, extensive research has been carried out, in order to explain the role played by TDs as nonradiative recombination centers (NRCs) [7,8]. The relative insensitivity of InGaN/GaN QWs to TD has been explained by a two-fold consideration: (i) carrier diffusion in QWs is quenched due to random alloy potential fluctuations that induce carrier localization [9,10], and (ii) TDs induce V-pits at the growing surface, which in turn lead to the formation in the QWs of a potential barrier around the TDs [11]. On the other hand, other defects, such as crystallographic point defects or impurities, may also degrade QW efficiency. Indeed, recent works have highlighted that point defects are also of great concern for the efficiency of InGaN/GaN QWs [12,13]. In particular, it was shown that defects lying at the GaN surface during the high-temperature (HT) buffer growth generate highly detrimental NRCs in InGaN/GaN QWs [13,14,15]. Those deep traps can be eliminated by introducing an InGaN (or InAlN) underlayer, which buries surface defects before they reach the QW active region [12,16,17]. These studies have shed light on the structure of high-efficiency blue LEDs, which always feature an InGaN layer or an InGaN/GaN superlattice underneath the active region.

In this work, we investigated the efficiency of single InGaN/GaN QWs with TD densities as high as 2–3 $\times \;10^{10}$ cm${}^{-2}$. This value is about one order of magnitude higher than the two-dimensional concentration of point defects in QWs associated with surface defects in the absence of an underlayer [13,15]. By varying the concentration of point defects in the QWs via the insertion of an underlayer, we show that they have a much more detrimental impact on QW efficiency than TDs, which were in our case efficiently screened by V-pits. This was confirmed by temperature-dependent time-resolved photoluminescence (TRPL) and cathodoluminescence (CL) experiments.

## 2. Materials and Methods

In this study, we prepared two samples grown on Si(111) substrates by metal-organic vapor-phase epitaxy (MOVPE) in an Aixtron 200/4 RF-S low-pressure horizontal reactor. The growth of the samples started with a 50 nm-thick AlN layer, followed by a 600 nm-thick GaN buffer. The active region consisted of a single 2.6 nm In${}_{0.17}$Ga${}_{0.83}$N QW sandwiched between two GaN layers of thickness 25 nm and 10 nm, respectively. The growth was terminated by a 5 nm-thick Al${}_{0.05}$Ga${}_{0.95}$N cap, to hinder surface recombination. The combination of a thin cap and a low acceleration voltage for the CL mappings enabled a high spatial resolution [18]. This reference sample was named R. All the layers were grown at a temperature of 750 °C, except for the AlN layer and the GaN buffer, which were grown at 970 °C, i.e., at HT. Sample U was similar to sample R, except for an In-containing underlayer introduced below the QW region, with the aim of reducing the point defect density [16,17]. This underlayer consisted of a 16-period lattice-matched In${}_{0.18}$Al${}_{0.82}$N(2.1 nm)/GaN(1.75 nm) superlattice. Such an underlayer has already been proven to be efficient at removing deep traps in InGaN/GaN QWs [17,18]. The structure of samples R and U is displayed in Figure 1a.

The surface morphology of the samples was assessed by scanning electron microscopy (SEM) and the structural properties by transmission electron microscopy (TEM). Prior to analysis with an FEI Talos F200S TEM, ∼180 nm-thick sample lamellas were prepared, parallel to the $\left(1\bar{1}00\right)$ crystallographic plane, by focused ion beam lift-out, using a Zeiss NVision 40 instrument. Power- and temperature-dependent photoluminescence (PL) studies were carried out, using quasi-resonant excitation with a continuous wave (cw) semiconductor laser diode emitting at a wavelength of 375 nm. The laser beam was focused by means of a near-UV 100 × Mitutoyo microscope objective, to reach a spot diameter of ∼1 µm. The PL signal was collected, using the same objective, and it was directed into an iHR320 Horiba monochromator equipped with a Peltier-cooled charge-coupled device camera. Spatial filtering was implemented on the collection path, to select only the light coming from the central part of the excitation spot, which corresponded to homogeneous in-plane carrier density (see Appendix A). TRPL was performed, using quasi-resonant excitation with a pulsed semiconductor laser emitting 40 ps pulses at a wavelength of 375 nm with a 500 kHz repetition rate. The beam diameter was 3 µm. CL mapping was performed with an Attolight Rosa 4364 system, using an acceleration voltage of 2 kV to ensure maximal resolution; such a voltage caused the energy to be absorbed mainly in the first 20 nm of the samples [18], which was the reason why the QW was located 15 nm under the surface (Figure 1a).

## 3. Results and Discussion

### 3.1. Morphological Study

Figure 2a,b display the surface of the two samples, as observed by SEM. They both exhibit a similar density of V-pits, ($n_{V-\mathrm{pit}}$), which are supposed to form at the termination of TDs [19]. The pit density and their corresponding average diameter ($d_{V-\mathrm{pit}}$) are reported in Table 1. To account for local variations, these values were calculated by averaging image-processed data over five areas of ∼1 × 1 µm${}^{2}$. The estimated V-pit density was about $3\times 10^{10}$ cm${}^{-2}$ for both samples. Most of them were distributed along loops, which corresponded to grain boundaries. [20] As already mentioned, V-pits are likely induced by TDs and, therefore, their density should be strongly correlated to the TD density. To confirm the relation between V-pits and TD densities, we performed cross-sectional TEM measurements. Proper diffraction conditions allowed us to discern either screw TDs, with diffraction vector g=(0002) parallel to the TD Burgers vector, or edge ones under a $g=(11\bar{2}0)$ condition. Mixed TDs were visible under both conditions [21]. For each sample, we displayed weak-beam dark-field images in the two different diffraction conditions (Figure 2c–f). The TD densities estimated from the statistical analysis performed on these TEM images are reported in Table 1. The total TD density amounted to $2.3\times 10^{10}$ cm${}^{-2}$ and $2.1\times 10^{10}$ cm${}^{-2}$ for samples R and U, respectively. These values were slightly lower than the V-pit density measured from the SEM images, likely due to the overlay of TDs in the TEM lamellas. We can conclude that the $n_{V-\mathrm{pit}}$ value measured in each sample was representative of the actual density of TDs ($n_{\mathrm{TD}}$), in agreement with previous observations done on samples grown on sapphire substrate [22]. For both samples, we found that nearly half of the TDs were pure edge, while pure screw TDs were much less present (see Table 1). Sample U was characterized by larger V-pits (41 nm in diameter) compared to sample R (20 nm), due to the growth of the underlayer at low temperature [23]. To summarize, the two samples exhibited a very high density of TDs, in excess of $2\times 10^{10}$ cm${}^{-2}$.

### 3.2. Optical Measurements

In order to compare the QW emission efficiency of these samples, we performed PL measurements as a function of excitation irradiance ($\Pi_{\mathrm{exc}}$) from cryogenic (8 K) to room temperature. Both samples had their QW PL spectrum centered at an energy comprised between 2.6 and 2.7 eV at 300 K, as shown in Figure 1b. From the power-dependent dataset, we calculated the ratio of the integrated intensity of the QW PL signal ($I_{\mathrm{PL}}$) over $\Pi_{\mathrm{exc}}$, and we display it as a function of $\Pi_{\mathrm{exc}}$ (Figure 3).

At this stage, we emphasize that both samples presented a total thickness comparable to the PL emission wavelength. Hence, instead of exhibiting a conventional Lambertian emission pattern, these samples were subjected to microcavity effects that induced significant changes in the light extraction efficiency from sample to sample, as explained in Appendix A. This prevented us from directly comparing their absolute PL intensity. We thus defined $\Pi_{\max}$ as the irradiance value at which $I_{\mathrm{PL}}/\Pi_{\mathrm{exc}}$ reached its maximum. At $\Pi_{\max}$, the internal quantum efficiency (IQE) reached its maximum, and the lower the $\Pi_{\max}$, the higher the absolute IQE [24]. By comparing the main features of samples R (without underlayer) and U (with underlayer) at T= 8 K, one can see that both samples exhibited $\Pi_{\max}$ spreading over a plateau centered at an irradiance near $1\times 10^{3}$ W cm${}^{-2}$, which testified to a rather high IQE, as expected at low temperature (Figure 3). Upon increasing the temperature up to T= 290 K, $\Pi_{\max}$ for sample U slightly increased to $3\times 10^{3}$ W cm${}^{-2}$, whereas the increase in $\Pi_{\max}$ was much larger, by two orders of magnitude, for sample R ($3\times 10^{5}$ W cm${}^{-2}$), indicating a drastic reduction of its IQE [24]. On the other hand, the introduction of an underlayer clearly allowed for sustaining high efficiency until room-temperature. The lower efficiency of sample R at room temperature could be explained by the smaller V-pits compared to sample U, which would have reduced the energy barrier surrounding the TDs [11]. To get a deeper insight into the impact of the V-pit size, we grew a control sample with similar point defect density as R (without underlayer) but with larger V-pits (see Appendix A). This was achieved by growing a thin GaN interlayer at low temperature underneath the QW active region [23]. Interestingly, while the V-pit diameter was larger (40 nm), the temperature dependence of the QW efficiency was very close to that of sample R (see Appendix A). Therefore, the low efficiency of sample R could not be ascribed to smaller V-pits and must, instead, have been due to the absence of an underlayer. This indicates that the insertion of an underlayer still has a dramatic effect on the improvement of QW efficiency, even for a TD density in the $10^{10}$ cm${}^{-2}$ range. Also, the moderate decrease of the IQE of sample U, upon increasing temperature, speaks for the marginal impact of TDs. Thus, the strong reduction of the QW efficiency of sample R seems to have been mainly due to the thermal activation of nonradiative channels related to point defects and to the increase of carrier diffusion length following their delocalization above a certain temperature (T∼ 50–70 K) (see Appendix A).

In order to further build upon the previous cw PL experiments, we performed complementary temperature-dependent TRPL measurements, to correlate the evolution of the relative IQE with the effective carrier lifetime ($\tau_{\mathrm{eff}}$). The lifetime values were extracted from TRPL decay curves at early delays, for different temperatures within the range T= 20–290 K (Appendix A). While the role of NRCs is often considered to be negligible at cryogenic temperatures (i.e., $\tau_{\mathrm{NR}}\gg \tau_{R}$, where $\tau_{\mathrm{NR}}$ and $\tau_{R}$ are the carrier nonradiative and radiative lifetimes, respectively), their impact becomes more significant with increasing temperature. This was the case for sample R, for which $\tau_{\mathrm{eff}}$ reduced from 54 ns at 20 K to 2 ns at 290 K. On the other hand, $\tau_{\mathrm{eff}}$ merely decreased from 75 ns at 20 K to 34 ns at 290 K for sample U. Note that the shorter effective decay time for sample R at 20 K indicates that NRCs are still active at low temperature, as already pointed out in Ref. [18]. In summary, the TRPL results strongly support the conclusion deduced from PL spectroscopy. In particular, the rather long TRPL decay time (>30 ns) measured at room temperature on sample U confirmed the low activity of TDs, despite their very high density ($2.1\times 10^{10}$ cm${}^{-2}$).

### 3.3. Cathodoluminescence Measurements

With the aim of gathering information at the nanoscale on the emission properties, we carried out CL mappings. At each excitation point, electron-hole pairs stood in an excitation volume with a typical diameter of 50 nm (see Appendix A). The CL setup also embedded a secondary electron (SE) detector, which allowed for comparing the surface morphology to the emission patterns. After fast relaxation into the QW, the carriers diffused over lengths of typically $L_{d}=2\sqrt{D\tau_{\mathrm{eff}}}$, with *D* the temperature-dependent diffusion coefficient. At room temperature, experimental *D* values for comparable In${}_{x}$Ga${}_{1-x}$N QWs varied from 0.27 cm${}^{2}$s${}^{-1}$ (x= 0.23) [25] to 6 cm${}^{2}$s${}^{-1}$ (x= 0.13) [26], and tended toward zero at low temperature. For sample U (respectively, R), this would correspond to $L_{d,U}>$ 1.9 µm (resp., $L_{d,R}>$ 0.5 µm) at room temperature. One should note that large diffusion lengths (>1 µm) have been recently reported in high-quality InGaN/GaN QWs grown on GaN substrates [26,27].

Figure 4a,b display panchromatic CL images. Figure 4a (sample R) exhibits areas of different CL intensities with a typical size of 0.5 µm in diameter. These domains correspond to crystallographic grains, where the boundaries are revealed by the V-pits (Figure 4c). The large intensity fluctuations indicate that the IQE is not uniform across the sample. Also, the sharp contrast variation is not compatible with a diffusion length > 0.5 µm. The intensity domains seem independent of each other. We propose that grain boundaries form domain walls for carrier diffusion mainly due to the potential energy barrier induced by V-pits [28,29,30] and local step bunching between the pits. Atomic force microscopy images (see Appendix A) reveal strong local surface misorientation, which, in turn, is known to reduce In incorporation [31]. Therefore, the QW energy locally increases, forming a barrier for carrier diffusion. We point out that the domain size is comparable to the carrier diffusion length, which explains the homogeneous intensity observed in each domain. To explain the CL intensity variations seen in Figure 4a, we assumed that each domain contained a different number of point defects. To estimate this number, we considered a point defect concentration of ∼1 $\times \;10^{16}$ cm${}^{-3}$, inferred from previous studies on InGaN QWs [15,16,18]. We point out that such structures are usually grown on sapphire or free-standing GaN, i.e., for different strain states and TD densities, which can lead to slight differences in the point defect density. This value was in line with another recent study highlighting the predominant role of the GaN buffer growth temperature, which is commonly ∼1000 °C for MOVPE growth [14]. In a 2.6 nm InGaN QW, such a density would correspond to about 5 defects over an area of ∼0.5 µm in diameter (i.e., a surface density of ∼2.6 $\times \;10^{9}$ cm${}^{-2}$). Statistical distribution of those point defects could thus be responsible for the different CL intensities of the domains. Potentially, the very bright domains could be free of any point defects, hence exhibiting a very high IQE. The CL mapping pattern shows that the IQE was not a macroscopic property in highly defective InGaN QWs but strongly varied across the sample at a submicron scale. In contrast, sample U displayed a rather homogeneous CL intensity, except for the presence of dark spots ascribed to V-pits. From our previous studies, the point defect density was ∼1 $\times \;10^{15}$ cm${}^{-3}$ in the present sample, which corresponded to around 20 point defects over the whole CL mapping area [15,18]. On average, half of the domains were point defect free, while the other half contained only one. The difference was hardly distinguishable on the CL intensity mapping, due to the full-scale range of the intensity. To be more quantitative, we extracted intensity histograms for both normalized CL intensity mappings (Figure 4e,f). Sample R exhibited a peak at low intensity (0.25), in agreement with a predominant occurrence of domains with the maximum number of point defects (5). Interestingly, the broadening of the peak could be attributed to domains with more or fewer point defects due to statistical fluctuations. In stark contrast, sample U had a broader peak at higher intensities (spreading from 0.7 to 0.85). The shape difference with the histogram of sample R can be tentatively ascribed to a binary distribution of point defect number per domain, namely, 0 or 1. Note that the interpretation we propose is in line with previous reports. Ding et al. observed similar CL intensity fluctuations and attributed them to gross well width fluctuations yielding barriers for carrier diffusion [32,33,34].

## 4. Conclusions

To conclude, this study reveals that in samples where the TD density is as high as 2–3 × 10${}^{10}$ cm${}^{-2}$, a strong increase in the InGaN/GaN QW quantum efficiency is observed upon introducing an underlayer, i.e., suppressing point defects originating from the HT growth of the GaN buffer layer. The corresponding long QW decay time (34 ns at 300 K) indicates that TDs have a weak nonradiative impact on the efficiency. This is consistent with a screening mechanism induced by V-pits, as indicated by SEM and CL imaging. On the other hand, the introduction of point defects in the QW, with a density of 2–3 × 10${}^{9}$ cm${}^{-2}$, results in dramatic decrease of the efficiency, which speaks for NRCs with a large cross-section. In addition, crystallographic grain boundaries lead to the formation of independent QW domains, which are delimited by V-pits and step edge bunching inducing potential barriers to carrier diffusion. This is evidenced by CL mappings showing strong intensity fluctuations across these domains. This is ascribed to the statistical distribution of the point defect number in each domain. Consequently, the macroscopic optical properties of such InGaN/GaN QWs should be interpreted, considering them as the sum of independent fractional QWs.

## Figures and Tables

**Figure 1 nanomaterials-13-02569-f001:**
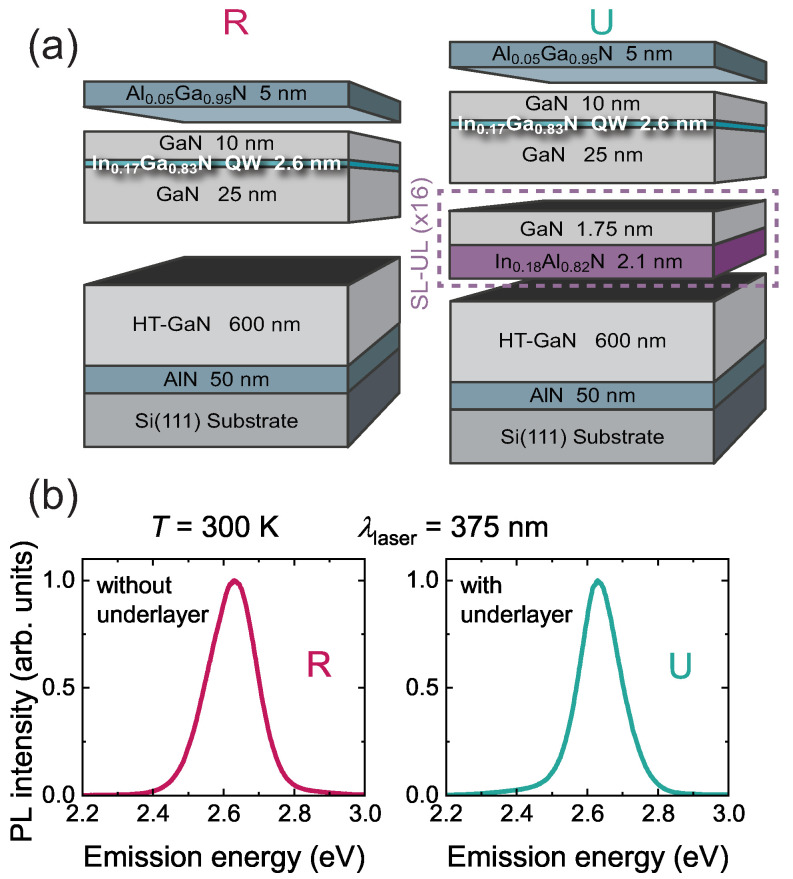
(**a**) Schematics of the investigated samples: R stands for reference sample, while U contains an underlayer; (**b**) room temperature QW PL spectra measured in the low power density regime. The intensity was normalized and the spectra were recorded from the samples’ edge, to eliminate cavity effects.

**Figure 2 nanomaterials-13-02569-f002:**
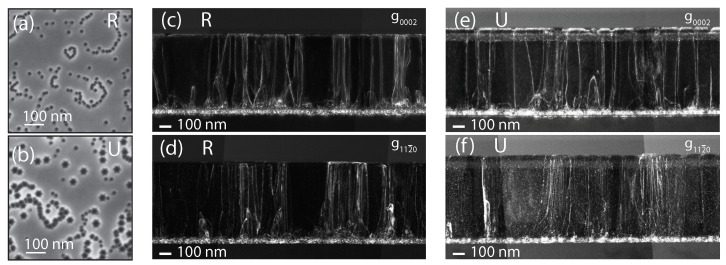
(**a**,**b**) SEM micrographs acquired at an accelerating voltage of 3 kV. The dark spots correspond to V-pits that formed at the termination of TDs; (**c**–**f**) cross-sectional TEM images of samples R (**c**,**d**) and U (**e**,**f**) with g=(0002) (**top row**) and $g=(11\bar{2}0)$ (**bottom row**) diffraction vectors.

**Figure 3 nanomaterials-13-02569-f003:**
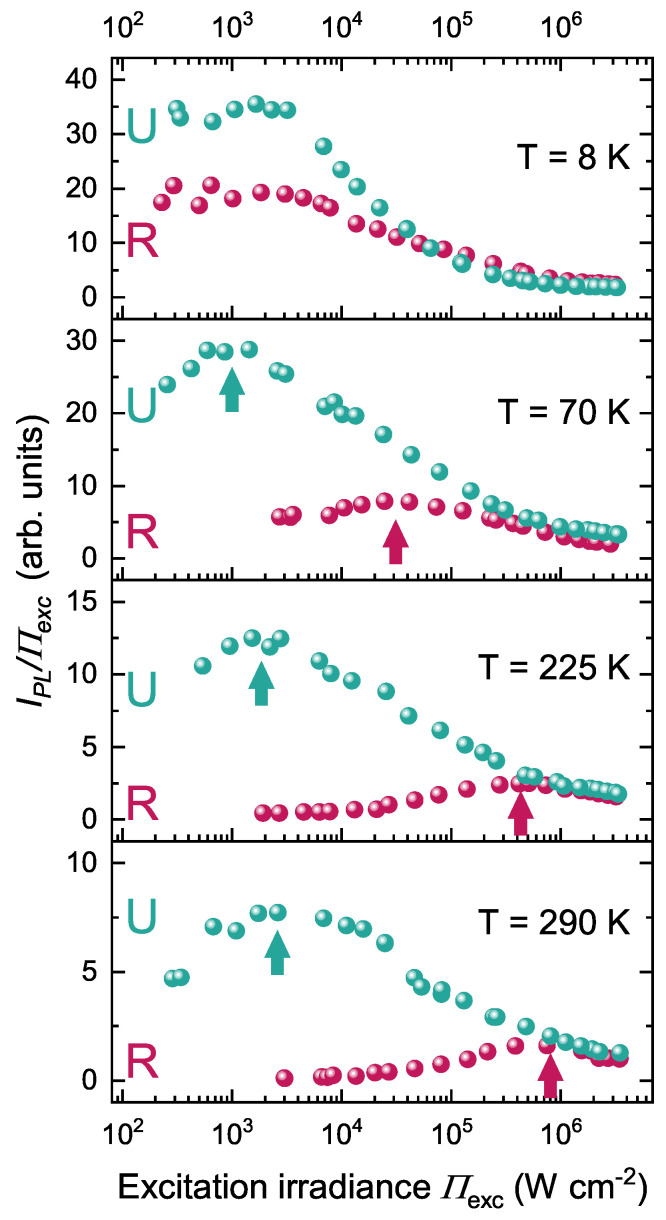
$I_{\mathrm{PL}}/\Pi_{\mathrm{exc}}$ as a function of $\Pi_{\mathrm{exc}}$ for T= 8, 70, 225, and 290 K for samples R (without underlayer) and U (with underlayer). The arrows indicate $\Pi_{\max}$.

**Figure 4 nanomaterials-13-02569-f004:**
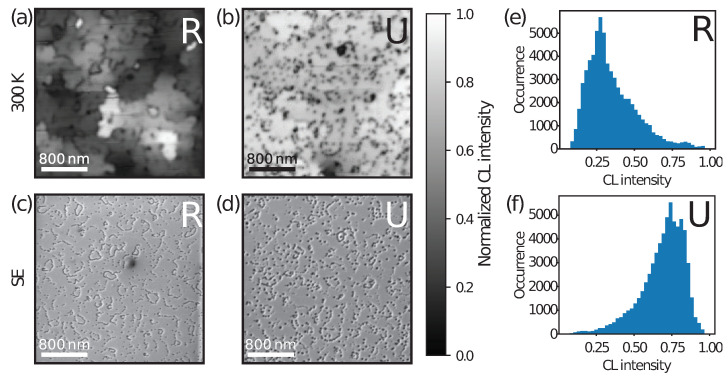
(**a**,**b**) show the panchromatic CL individually normalized integrated intensity images of 3.2 × 3.2 µm${}^{2}$ areas for samples R (**a**) and U (**b**), recorded at room temperature (groups of V-pits appear as large dark spots); (**c**,**d**) SE images for the same areas of each sample (R, U); V-shaped pits marking the position of TDs are visible as dark spots; (**e**,**f**) intensity histograms from the CL mappings.

**Table 1 nanomaterials-13-02569-t001:** V-pit diameter and density, TD densities, and dislocation nature, assessed from top-view SEM and cross-sectional TEM analyses.

Sample	SEM		TEM			
	V-pit Diameter (nm)	V-pit Density (10^10^ cm^−2^)	TD Density (10^10^ cm^−2^)	Mixed	Screw	Edge
R	20±2	3.3±0.4	2.3	37%	18%	45%
U	41±5	2.5±0.3	2.1	44%	8%	48%

## Data Availability

Data are available at https://doi.org/10.5281/zenodo.8223563 (accessed on 15 September 2023).

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
