# Peer review of "Investigation of the Impact of Point Defects in InGaN/GaN Quantum Wells with High Dislocation Densities"

_nanomaterials, 2023, doi:10.3390/nano13182569_

Round 1

Reviewer 1 Report

This work fabricated InGaN/GaN quantum wells using by metal-organic vapor phase epitaxy method and studied their emission properties using PL, TRPL, and CL measurement. The results indicate the introduction of PDs in the QW dramatic decrease of the emission efficiency, and UL increase the efficiency.  The results and observed phenomena are well analyzed and explained. I recommend accepting this manuscript after a minor revision.

The electron diffraction pattern and high resolution TEM are helpful to determine structures and diffraction vectors of samples.

Author Response

We thank Reviewer 1 for the critical reading of the manuscript.

We tried to improve the introduction for better readability.

We adapted the references to reduce the self-citation rate below 15%.

Reviewer 2 Report

In this manuscript, the authors investigated the influence of the point defect (PD) density on the luminescence efficiency of the InGaN quantum with extremely high-density of threading dislocations. By means of temperature and excitation density dependent photoluminescence, temperature dependent time-resolved photoluminescence, and cathodoluminescence, they revealed that even in the sample with TDDs of up to the order of 10^10 cm^-2, PDs with the density of only 10^9 cm^-2 act as the dominant non-radiative recombination path. The findings shown in this manuscript shed light on the importance of the reduction of PDs in InGaN based light-emitting devices. The experiment is fully complemental and analyzed in depth. Thus, I recommend the publication of this manuscript in Nanomaterials just after the following minor revisions.

#1

In line 163, the authors say that “To estimate this number, we consider a PD concentration of ∼ 1×10^16 cm^−3.” I could not understand why this PD density was derived. I’d like to request the authors to add more detailed explanation.

Author Response

We thank Reviewer 2 for the careful reading of our work.

Reviewer 2's comment: "In line 163, the authors say that “To estimate this number, we consider a PD concentration of ∼ 1×10^16 cm^−3.” I could not understand why this PD density was derived. I’d like to request the authors to add more detailed explanation."

To tackle the imprecision about point defect densities, we explain more extensively how we inferred the given value in the text.

Reviewer 3 Report

In this study, the authors detail the impact of an underlayer on the efficiency of grown quantum wells. The authors show that there are significant differences, and the growth of the underlayer significantly improves their desirable properties. The authors also investigate the mechanism by which this underlayer improves the efficiency, attributing it to different types of defects in the material. Overall, this is a sound study and I find no significant errors or any major changes that should be made. I have two small comments that I believe would slightly improve the manuscript.

Small comment, but in Figure 1 I think that the figure caption should state that R is the reference and U contains the underlayer (I believe)? Just would be helpful for clarity.

I also think the abundance of acronyms impedes the readability of the manuscript. I would remove acronyms for threading dislocation, point defects, underlayer, superlattice, etc. It makes the manuscript more opaque for the reader as they have to remember what these acronyms stand for.

Author Response

We thank Reviewer 3 for the insightful comments.

First comment : "Small comment, but in Figure 1 I think that the figure caption should state that R is the reference and U contains the underlayer (I believe)? Just would be helpful for clarity."

In turn, we added the suggested information in Fig. 1.

Second comment : "I also think the abundance of acronyms impedes the readability of the manuscript. I would remove acronyms for threading dislocation, point defects, underlayer, superlattice, etc. It makes the manuscript more opaque for the reader as they have to remember what these acronyms stand for."

We also removed the lesser-known acronyms, i.e. “PD”, “UL” and “SL”.